# Oxidative Stress in Malaria: Potential Benefits of Antioxidant Therapy

**DOI:** 10.3390/ijms23115949

**Published:** 2022-05-25

**Authors:** Antonio Rafael Quadros Gomes, Natasha Cunha, Everton Luiz Pompeu Varela, Heliton Patrick Cordovil Brígido, Valdicley Vieira Vale, Maria Fâni Dolabela, Eliete Pereira de Carvalho, Sandro Percário

**Affiliations:** 1Post-Graduate Program in Pharmaceutica Innovation, Institute of Health Sciences, Federal University of Pará, Belém 66075-110, PA, Brazil; rafaelquadros13@hotmail.com (A.R.Q.G.); helitom2009@hotmail.com (H.P.C.B.); valdicleyvale@gmail.com (V.V.V.); fani@ufpa.br (M.F.D.); 2Oxidative Stress Research Laboratory, Institute of Biological Sciences, Federal University of Pará, Belém 66075-110, PA, Brazil; natasha_nx@hotmail.com (N.C.); evertonlpvarela@gmail.com (E.L.P.V.); elicar28@uol.com.br (E.P.d.C.); 3Post-graduate Program in Biodiversity and Biotechnology (BIONORTE), Institute of Biological Sciences, Federal University of Pará, Belém 66075-110, PA, Brazil

**Keywords:** oxidative stress, malaria, antioxidants, supplementation, free radicals

## Abstract

Malaria is an infectious disease and a serious public health problem in the world, with 3.3 billion people in endemic areas in 100 countries and about 200 million new cases each year, resulting in almost 1 million deaths in 2018. Although studies look for strategies to eradicate malaria, it is necessary to know more about its pathophysiology to understand the underlying mechanisms involved, particularly the redox balance, to guarantee success in combating this disease. In this review, we addressed the involvement of oxidative stress in malaria and the potential benefits of antioxidant supplementation as an adjuvant antimalarial therapy.

## 1. Introduction

Malaria is an infectious disease and a serious public health problem, with 3.3 billion exposed people in endemic regions at risk of contracting the disease in 85 countries. According to the latest World Malaria Report from the World Health Organization (WHO), in 2020, there were an estimated 241 million cases of malaria and 627,000 malaria deaths worldwide. Compared to the year 2019, there was an increase of about 14 million new cases and 69,000 more deaths. About 47,000 of these additional deaths were related to disruptions in the provision of malaria prevention, diagnosis, and treatment during the COVID-19 pandemic [1].

In the Americas, about 138 million people live in areas at risk of malaria, with 765,000 cases of the disease being reported [2]. Venezuela, Brazil, and Colombia are the countries with the highest number of cases and deaths resulting from the disease on this continent, contributing 86% of the cases [1].

To date, there are six species of *Plasmodium* reported to infect humans, however, the deadliest form of the disease, cerebral malaria, is associated with *Plasmodium falciparum* infection, although there are reports describing severe manifestations, and even deaths, caused by *Plasmodium vivax* [3]. The main factors that make malaria a worldwide issue that is difficult to eradicate are difficulties in accessing treatment, the migration of individuals to endemic areas, the lack of an efficient vaccine, the level of socioeconomic development in endemic areas, and, mainly, parasite resistance to antimalarials [4]. 

*Plasmodium* resistance to drugs is growing worldwide, especially in endemic areas. In addition to quinine and chloroquine [5], resistance has been reported to proguanil [6], atovacone [7], mefloquine [8], and to combination therapy with artemisinin [9,10]. In this context, for a better understanding of the resistance factors and pathophysiological mechanisms of the disease, it is necessary to deepen our knowledge of the underlying mechanisms of the disease, particularly regarding redox regulation in malaria.

Infection by the parasite triggers oxidative stress that consists of the host’s redox imbalance due to the high production of reactive oxygen and nitrogen species (ROS/RNS), to the detriment of the host’s endogenous antioxidant defenses. The production of these oxidizing compounds occurs through the degradation of hemoglobin by the parasite, which uses it as a source of amino acids for its nutrition, as well as through the host’s immune response, which produces ROS/RNS inside the phagocytes, a process called respiratory burst, alongside the involvement of ischemia-reperfusion syndrome (SIR) [11].

The increase in oxidative stress in malaria is responsible for systemic and tissue oxidative damage, affecting organs such as lungs and brain. This redox imbalance has been related to the most serious manifestations, such as cerebral malaria, and, depending on the lesions in the brain parenchyma, it can lead to physical and cognitive impairments [12].

To combat oxidative changes, the host organism has an antioxidant defense system that uses enzymes and small antioxidant molecules to fight free radicals produced by the infection [11]. However, those defenses are often insufficient to fulfill the host’s needs [13]. Therefore, studies have sought adjuvant antioxidant treatments to support the host’s antioxidant defense system [14,15,16,17].

Supplementation with antioxidants represents a promising therapeutic alternative in malaria, which would prevent or reduce the oxidative damage generated by the parasite and by antimalarials. Based on this idea, several studies have demonstrated the beneficial role of antioxidant supplementation in fighting the infection and restoring redox balance. The supplementation with vitamins A [18,19,20,21], C [17,22,23], D [15,24,25], and E [19,26] represent promising strategies, with possible beneficial effects on the disease.

Supplementation with other substances is also reported, such as N-acetylcysteine (NAC) [14,27], minerals such as zinc [18,28,29] and selenium [30,31,32], and mushrooms such as *Agaricus sylvaticus* [14].

In addition, it has been reported that vitamin A and zinc supplementation in pregnant women prevents placental malaria [29]. Additionally, therapy with the antioxidants NAC and tempol can improve pregnancy in animals. However, tempol has demonstrated higher antioxidant activity and, therefore, reduced oxidative and inflammatory damage in a more effective way, increasing embryo survival [33].

Thus, this review aims to demonstrate the role of oxidative stress in malaria, emphasizing the benefit of antioxidant supplementation as a complementary therapy to drug treatment.

## 2. Oxidative Stress in the Host, Induced by *Plasmodium*

Oxidative stress in malaria is involved in various stages of infection and, depending on the stage of the disease, can be both harmful and beneficial during plasmodial infection [34,35,36]. In fact, high levels of oxidative stress in the blood of patients with malaria have already been seen through the increase in malondialdehyde (MDA), a by-product of lipid peroxidation [37].

In a similar study, Nsiah et al. [38] identified increased levels of oxidative stress in children with complicated malaria infection, through increased levels of malondialdehyde (MDA), and decreased ascorbate and hemoglobin in children. A similar result was reported by Atiku et al. [34], in which severe oxidative stress was observed in patients with uncomplicated sickle cell disease and *P. falciparum* malaria.

Complications in severe malaria are systemic, with impairment to the pulmonary and central nervous system. Pulmonary manifestations such as cough with or without sputum and dyspnea have been described [39]. Another serious outcome is acute respiratory distress syndrome (ARDS), characterized by increased vascular permeability, resulting in pulmonary edema. In parallel, Anidi et al. [40] demonstrated that erythrocytes sequestered in the lung produce ROS/RNS and increase vascular permeability, which is exacerbated by the high expression of CD36 and Fyn kinase in *P. berghei* infection, suggesting them as mediators of the increase in the conductance of pulmonary endothelial fluid.

Cerebral malaria is a fatal complication, and oxidative stress in the central nervous system is related to the development and worsening of the disease [39]. Studies have demonstrated that oxidative damage caused by *P. berghei* infection in mice has been associated with brain damage, with long-term physical and cognitive dysfunction [41]. The parasite presence in the brain increases oxidative stress, since the adhesion of erythrocytes in vessels causes the obstruction of blood flow, resulting in ischemia and reperfusion syndrome. In the reoxygenation of the tissue, ROS/RNS are produced, as well as in the release of free heme due to the lysis of erythrocytes and in the activation of the immune response through the release of proinflammatory cytokines, such as interferon-ɣ (IFN-ɣ), tumor necrosis factor-α (TNF-α), and lymphotoxin-α (LT-α) by phagocytic cells in the respiratory burst [42]. These conditions in lung and brain tissue increase microvascular permeability, resulting in the pulmonary syndrome associated with severe malaria [40].

In addition to the involvement of these key organs, the role of oxidative stress in the myocardial wall and in the high pulmonary pressure in children with severe malaria has also been demonstrated, and it is caused by intravascular hemolysis and a decrease in nitric oxide (NO) production, with a consequent cardiopulmonary effect [43]. Additionally, Na-Ek and Punsawad [44] identified increased expression of 4-hydroxynonenal (4-HNE) and heme oxygenase-1 (HO-1) in the renal tissue of mice, both markers of oxidative damage and related to acute tubular damage in the kidneys during experimental malaria.

According to Percário et al. [11], the induction of oxidative stress in malaria is multifactorial: the host’s inflammatory response triggered by the parasite; transition-metal-induced free radical production derived from the consumption of hemoglobin (through Fenton and Haber–Weiss reactions); ischemia and reperfusion syndrome, caused by cytoadherence in the microvasculature and anemia during infection; direct generation of free radicals induced by the parasite; and the use of antimalarial drugs (Figure 1). 

In the initial stage of infection, neutrophils are one of the first defense cells. When activated, they can destroy malaria parasites through various mechanisms, mainly through the production of ROS/RNS through oxidative stress [45]. The neutrophils of patients with malaria manifest differently from those who are not infected [46], and neutrophils in patients infected with *P. falciparum* present impaired oxidative burst due to increased plasma heme, resulting in cerebral vasculopathy [47]. Additionally, the actions of other immune cells during infection are reported, such as macrophages, lymphocytes, and dendritic cells [48,49,50,51,52,53,54].

The proinflammatory response is associated with the severity of cerebral malaria, with an increase in nitric oxide (NO) and hydrogen peroxide (H_2_O_2_), but the regulation of these factors is still unclear. According to Borges et al. [55], the expression of eicosanoid-producing enzymes in mice sensitive to cerebral malaria increases the expression of cyclooxygenase-2 (COX-2) and 5-lipoxygenase (5-LOX) in the vessels and brain tissue, followed by high levels of the transcriptional regulators of lipid metabolism, peroxisome proliferator-activated receptor gamma (PPAR-γ), which results in increased parasitemia, reduced survival, and less NO and H_2_O_2_ production in those animals.

During the inflammatory process of the disease, oxidative stress influences the maturation and function of dendritic cells (DC) in response to infection by *P. falciparum*, using xanthine oxidase (XO), and the ROS/RNS generated by this enzyme increase the secretion of cytokines induced by the parasite and the surface expression of CD80 in DC. Therefore, oxidative stress during infection contributes to the inflammatory response, with an increase in CD4^+^ T cells and DC [35,51]. It is worth mentioning that the activation of CD4^+^ T cells by DC give rise to subsets of helper T cells (Th), including Th1, Th2, or Th17, leading to the production of IFN-γ, interleukin-4 (IL-4), or IL-17, respectively [56].

This suggests that the severity of infection and inflammatory cytokine production correlate with plasma XO levels in patients with malaria. In a study conducted by Ty et al. [57], the ROS/RNS produced by XO-induced inflammatory cytokines in macrophages and correlated with the development of cerebral malaria. In addition, the reactive species produced also promote the production of pro-IL-1β, while erythrocytes infected by the parasite activate the inflammasome NOD-like receptor family pyrin domain-containing 3 (NLRP3), which acts on the activation of caspase 1, and the consequent cleavage and release of bioactive IL-1β. 

Malarial infection results in a high production of ROS/RNS, which are harmful to both the host and the parasite. Initially, the parasite inside the erythrocyte degrades hemoglobin to use its amino acids for its nutrition, and the rapid growth of the parasite in the host cell is not only dependent on nutritional absorption but also on the redox balance [58]. The iron of the heme group released from hemoglobin reacts with H_2_O_2_ through the Fenton (or Haber–Weiss) reaction, oxidizing the iron and producing highly toxic free radicals, such as the hydroxyl radical (OH^•^) and superoxide (O_2_^●--^) [58,59]. In this process, the organism increases the expression of superoxide dismutase (SOD) to detoxify the microenvironment of the production of O_2_^●-^, generating in this reaction even more H_2_O_2_ and OH^•^ [60].

The oxidative stress generated at the erythrocytic level affects both the parasite and the host cell through oxidative changes triggered mainly by the lipid peroxidation of red blood cells [61,62]. However, to protect itself from free radicals, the parasite uses its apicoplast organelle to produce the antioxidant lipoic acid as a defense mechanism [63]. Burda et al. [58] identified by means of X-ray crystallography a protein of lipocalin monomers called *P. falciparum* lipocalin (PfLCN) expressed during the parasite intra-erythrocyte development. It is located in the parasitophoric and digestive vacuoles, neutralizing the cellular damage caused by oxidative stress during the reproduction of *Plasmodium* sp. in erythrocytes.

Moreover, due to the high density of parasitized erythrocytes in the pulmonary and cerebral microvasculature obstructing blood flow, ischemia and reperfusion syndrome occurs, producing hypoxia in these tissues, which is probably responsible for pulmonary and brain lesions. Since free radicals and inflammatory components are generated in this syndrome, the damage is accentuated [11,64,65].

Besides host ROS/RNS production in response to infection, the parasite itself is capable of producing free radicals, which in turn interfere with the biochemistry of red blood cells and may promote or facilitate the internalization of the parasite in hepatocytes and red blood cells (RBC). Despite scarce exploration in the scientific literature, aerobic membrane transport mechanisms are a major source of ROS/RNS generation in *Plasmodium*. Accordingly, a recent study found that the absence of NADPH-oxidase expression, an important enzyme in the synthesis of free radicals by macrophages, caused no differences in the progression of parasitemia in knockout mice for any of the *Plasmodium* species tested *(P. yoelii*, *P. chabaudi* K562, *P. berguei* ANKA, *P. berguei* K173, and *P. vinckei vinckei)*. These findings led the authors to believe that free radical production increased as a result of infection and not from the respiratory burst of phagocytes, possibly due to production by the parasite itself [66]. Another factor that reinforces this possibility lies in the complexity and variety of antioxidant mechanisms developed by these parasites. 

Another factor that generates oxidative stress during plasmodial infection is the treatment itself, as antimalarial drugs are pro-oxidant sources [67,68]. An example is mefloquine, which has a blood schizonticide effect and, by interfering with the heme detoxification process, produces ROS/RNS against the parasite [69,70]. An in vitro study also demonstrated that mefloquine acts by binding to cysteine proteases, causing parasite apoptosis mediated by ROS/RNS production [71].

In a study by Giovanella et al. [72], in which the effects of primaquine and chloroquine on oxidative stress and DNA damage in mice were evaluated, it was identified that primaquine caused DNA damage in brain and liver, while chloroquine caused damage to kidneys, liver, and brain, with increased MDA levels in both cases, suggesting that, during the treatment, these drugs induced oxidative stress and DNA damage.

Artemisinin and its derivatives also display antimalarial activity by inducing ROS/RNS production, which oxidizes proteins and lipids within infected erythrocytes [73]. Furthermore, artemisinin and its derivatives, dihydroartemisinin and artesunate, act by causing the rapid depolarization of the parasite’s membrane potential, which can be inhibited by antioxidants and iron chelators [74]. An in vitro study demonstrated the blockage of heme polymerization using artemisinin-heme adducts and *Plasmodium*-rich histidine-II and III proteins, generating an accumulation of toxic ferriprotoporphyrin IX, with consequent oxidative stress [75,76]. Artemisinin-based combination therapy also involves the production of ROS/RNS [77].

### 2.1. The Role of Pro-Oxidants and NO in Malaria

Biological systems produce, through fundamental chemical processes for cell maintenance, unstable and highly reactive molecules that can cause oxidative damage. These molecules, called free radicals or ROS/RNS, are produced continuously by the cells and are involved in energy production, growth regulation, cell signaling, endothelial pressure control, and the phagocytosis of pathogens [78,79]. Among the free radicals and the most relevant reactive species are OH^●^, O_2_^●-^, NO, lipid peroxide radical (LOO•), H_2_O_2_, and singlet oxygen (^*^O_2_) [80,81,82].

Notwithstanding the harmful effects of ROS/RNS, their production can also occur from endogenous sources, for example, NADPH, myeloperoxidase (MPO), phagocytosis [83], and exogenously from pollution [84], smoking [85], alcohol use [86,87], transition metal poisoning [88], radiation exposure, nutritional deficiencies, oxidant drug metabolism, and medicines [89,90], as well as from pathogens [11,14,41]. These molecules, when present in high quantities in the organism, produce cellular damage when reacting with biomolecules, leading to deleterious cellular and tissue effects.

During malaria infection, ROS/RNS are involved in the destruction of *Plasmodium* via oxidative stress during the erythrocytic phase [91]. However, the early increase of these molecules induces oxidative stress via the production of extracellular methemoglobin, causing osmotic fragility and hemolysis [92]. The stimulation of ROS/RNS by neutrophils, monocytes, and macrophages is one of the main mechanisms of host defense against the parasite, generating an imbalance between pro- and antioxidant elements, triggering oxidative stress [93]. On the other hand, the mechanisms of immune response can also act to damage the host cells [11].

Defense cells such as neutrophils can destroy the parasite through respiratory explosion, initially producing a high amount of O_2_^●−^ from NAPH oxidase (NOX), acting on O_2_, which is later converted into H_2_O_2_ and OH^●^. The NOX enzyme is found in the cytoplasm and phagosome membranes of neutrophils, producing ROS/RNS that diffuse through the membrane, destroying extracellular phagocyted and intracellular parasites via oxidative stress [45].

One of the main ROS/RNS involved during malaria infection is NO, which is synthesized from the amino acid L-arginine, through the action of nitric oxide synthase (NOS). Three isoforms of NOS are characterized, according to tissue location, two constitutive forms (cNOS) and one inducible (iNOS). Constitutive forms include neuronal NOS or NOS-I, expressed in neuron cells during neurotransmission, and endothelial NOS or NOS-III, produced in endothelial cells to promote vasodilation. The inducible form (iNOS or NOS-II) is activated by inflammatory cells producing large amounts of NO in a short time [94,95].

The role of NO in malaria is still unclear, but some researchers suggest that cerebral malaria is related to high NO production to promote parasite death [95,96,97,98]. On the other hand, some support the idea that cerebral malaria results from the low bioavailability of this substance [99,100,101,102,103,104].

Indeed, NO can act in two main ways to control malaria, directly through the parasiticidal action of peroxynitrite generated by the reaction of NO with O2^●-^, inducing oxidative stress, or indirectly by increasing the immunological response via cytokine stimulation and increased ROS/RNS [11].

Moreover, many authors believe that NO and carbon monoxide (CO) display a protective effect against cerebral malaria, as they inhibit severe forms of the disease. This effect is related to three main mechanisms, which are the expression of erythroid-related nuclear transcription factor 2 (NRF-2), the induction of HO-1, and the production of CO via heme catabolism by HO-1.

In addition, the protective effect of NO on this disease is related to the inhibition of CD4^+^ and the activation of CD8^+^ T lymphocytes, through a mechanism related to HO-1 and CO [97]. Additionally, NOS II polymorphism increases the bioavailability of NO and protects from cerebral malaria [95]. In a similar study conducted by Lwanira et al. [105], the presence of a polymorphism in the NOS II gene resulted in fewer cases of malaria, as well as in children with normal hemoglobin.

However, there are researchers who believe that NOS polymorphisms do not prevent the onset of cerebral malaria. To them, this conclusion was reached from the evaluation of the genetic regulation of NO synthesis, through the levels of oxidative stress and polymorphisms of iNOS in *P. vivax* and *P. falciparum* malaria. These authors concluded that the iNOS polymorphism acts in the regulation of NO expression, and genotyping increases the risk of developing the disease, since there was a reduction in total antioxidant capacity and ROS during infection [106].

Nevertheless, there are studies that suggest that the reduction of NO bioavailability is related to the worsening of malaria, such as the study by Yeo et al. [101], in which the decrease in NO bioavailability in children with *P. falciparum* malaria compromised the microvasculature and resulted in greater consumption of O_2_ in the tissues. In addition, the reduced bioavailability of arginine and NO result in endothelial dysfunction in *P. vivax* malaria, causing microvascular injury [102].

The dysfunction of NOS is also related to the reduction of NO bioavailability in malaria, as in Ong et al. [100], who assessed the cerebrovascular capacity and the function of NOS isoforms in cerebral malaria in animals; the loss of eNOS and NOS-I isoforms functionality contributes to cerebrovascular injury, which is characterized by vascular constriction, impaired perfusion, and reduced cerebral blood flow, requiring the recovery of enzymes to increase NO bioavailability.

Another factor related to the reduction of NO is that *P. falciparum*-generated hemozoin, from the digestion of hemoglobin in infected red blood cells and released into the bloodstream, accumulates in leukocytes, and high levels are related to the severity of the disease. Studies have shown the low bioavailability of L-arginine and NO in *P. falciparum* malaria and positively correlated it with hemozoin levels [107,108]. In an in vitro study by Corbett et al. [109], *P. falciparum*-induced hemozoin reduced the bioavailability of the amino acid L-arginine to iNOS, with a consequent reduction in NO synthesis in macrophages. Moreover, monocyte polarization during *P. falciparum* malaria infection reduced the NO bioavailability and increased the severity of the disease in children [103].

Other authors support the idea that the reduction of NO bioavailability to a certain level in malaria by exogenous inhibitory substances used in the treatment provides a protective role against the worsening of the disease. This fact was identified by Martins et al. [100], who evaluated the efficacy of different treatments on NO synthesis in *P. berghei*-induced cerebral malaria in mice and observed that sildenafil (a phosphodiesterase-5 inhibitor) reduced the level of NO required to prevent cerebral malaria, while L-arginine and tetrahydrobiopterin increased NO bioavailability but did not increase the survival rate of infected animals.

According to Moreira et al. [110], the inhibition of cerebral NO synthesis by dexamethasone increased the survival rate in 90% of animals until the 15th day of infection, followed by a significant reduction in parasitemia in mice infected with *P. berghei.* On the other hand, when NO synthesis is inhibited by Nω-nitro-L-arginine methyl ester (L-NAME), there is an increase in animal mortality, with a consequent reduction in parasitemia in *P. berghei*-infected mice, suggesting that the physiological effects of NO outweigh its pro-oxidant role in experimental malaria [111].

In a similar study, the inhibition of NO synthesis by aminoguanidine in avian malaria caused by *P. gallinaceum* increased chickens’ survival, but there was an increase in parasitemia. Nevertheless, aminoguanidine was able to reduce thrombocytopenia, anemia, and inflammation by reducing hemozoin in the liver and spleen of infected animals [112].

Weinberg et al. [113], by inhibiting NO synthesis through asymmetric dimethylarginine in patients with *P. falciparum* malaria, identified that there is a reduction in arginine levels in both children and adults. It has been described that the reduction of liver activity by asymmetric dimethylarginine inhibits NO synthesis in severe malaria [114]. Based on this finding, the authors suggest that adjuvant therapies that increase the bioavailability of NO may be advantageous against the disease.

In this sense, to increase the bioavailability of NO in malaria, some researchers have suggested that supplementation [115,116] or infusion of L-arginine [117] and gaseous therapies based mainly on NO inhalation improve endothelial dysfunction and immunomodulate a host’s response to infection [104,118,119].

In a study by Zhu et al. [116], supplementation with L-arginine during infection by *P. yoelii* in mice reduced parasitemia and prolonged the survival rate of infected animals. In addition, there is an increase in activated CD4^+^ and CD36^+^ T cells and macrophages during the first stages of infection, followed by an increase in IFN-γ, TNF-α, and NO in spleen cells.

In addition to supplementation, the inhalation of NO gas improves clinical features in children with severe malaria who are undergoing standard antimalarial treatment, since this suggestion is based on NO acting on the modulation of endothelial activation, and supplementation with L-arginine is effective against cerebral malaria in animals and improves endothelial function [120]. A study on NO inhalation in children with severe malaria in Uganda identified a reduction in motor impairment, suggesting its neuroprotective role in plasmodial infection [104]. Thus, therapy through the inhalation of NO would represent an adjuvant alternative to treat severe malaria, with a possible reduction in mortality and neurocognitive problems [118].

Notwithstanding, other studies suggest that arginine therapy is controversial because the parasite has highly active arginase and can cause hypoarginemia in the host, impairing the bioavailability of NO [101,121,122]. Moreover, the administration of low doses of arginine did not restore the bioavailability of NO in patients with severe malaria [123]. However, in an experimental study in mice, arginine supplementation was able to increase NO levels but was not able to reduce the incidence of cerebral malaria [99]. Additionally, the parasite cannot consume citrulline, which could be a better therapeutic alternative to arginine supplementation, as it is a better NO donor than arginine, and, when administered orally, inhibits arginase better than the administration of arginine, protecting mice from cerebral malaria and improving hypoargininemia, urea cycle changes, and vascular leakage [122]. In addition, the beneficial effect of citrulline in inhibiting in vitro replication of *P. falciparum* has been reported [124].

### 2.2. The Host’s Antioxidant Defense Mechanism against Plasmodium

Any substance that, when present in low concentration compared to an oxidizable substrate, can prevent or delay the oxidation of the substrate is considered an antioxidant. Antioxidants are mainly involved in preventing, blocking, or quenching ROS/RNS and, consequently, oxidative stress, as well as other parameters of cell damage [125,126]. Endogenously produced antioxidants are classified as enzymatic or non-enzymatic.

During the infectious process of malaria, *Plasmodium* destroys red blood cells, leading to the production of large amounts of free radicals. In addition, neutrophils and macrophages are recognized for producing O2^●−^ radicals and H_2_O_2_, which are essential for defense against phagocytes or invasive parasites. Therefore, antioxidants are necessary to regulate the reactions that release ROS/RNS.

The main enzymatic antioxidants that constitute an important line of defense for the body include SOD, catalase (CAT), glutathione peroxidase (GSH-Px), glutathione-S-transferase (GST), and peroxiredoxins (Prx; Figure 2). These antioxidants can prevent cell damage by free radicals and are essential for the maintenance of optimal health in animals and humans [16].

There is enough evidence that antioxidants contribute to the balance between pro- and antioxidant status related to anemia development and clinical features of malaria infection [127,128]. In this sense, Farombi et al. [129] found a decrease in CAT and GSH-Px activity in patients with *P. falciparum* malaria.

CAT is a tetrameric hemoprotein present in liver cells and erythrocytes in high concentrations. It undergoes divalent oxidation and alternating reduction in its active site in the presence of H_2_O_2_, reducing it to water [75]. Like CAT, GSH-Px, glutathione reductase (GR), and Prx neutralize and break down H_2_O_2_ and lipid peroxides (ROOR) into less harmful molecules, such as H_2_O, alcohol, and O_2_ [16]. The thioredoxin-dependent system uses electrons provided by the thioredoxin (Trx) thiol groups to reduce its target, and the GST/GSH-Px system obtains electrons from reduced glutathione (GSH) [130].

However, the accumulation of H_2_O_2_ due to the decrease in CAT and GSH-Px activities in patients with malaria can inactivate SOD activity, which catalyzes the conversion of O_2_•^−^ into H_2_O_2_. In a study by Babalola et al. [131], reduced levels of SOD and GSH were identified, followed by an increase in MDA in patients with asymptomatic malaria. Moreover, Tyagi et al. [132] found a decrease in the activity of SOD, CAT, and GST, simultaneously with the increase in GSH and MDA in patients with untreated malaria and, when starting treatment with antimalarials, the inversion of this condition was observed i.e., the increased SOD, CAT, and GST activities, followed by the reduction of GSH and MDA. Similar results were found by Oluba [133], with a decrease in GSH, SOD, CAT, and GSH-Px in children with *P. falciparum* malaria, but the start of antimalarial treatment resulted in an increase in the activity of GSH, SOD, CAT, and GSH-Px.

In parallel, Ezzi et al. [13] demonstrated reduced levels of SOD in patients with malaria. For these authors, SOD plays an important role as an antioxidant, and decreased levels demonstrate the effort to compensate the oxidative stress associated with the disease. Furthermore, increased levels of ceruloplasmin were found in these patients. According to the authors, ceruloplasmin, which is a ferroxidase, was activated to catalyze the oxidation of ferrous iron (Fe^2+^) to ferric iron (Fe^3+^), preventing a Fenton reaction and the consequent production of OH^●^ radicals.

Previously, a clinical study conducted by Fabbri et al. [134] showed an increase in ceruloplasmin levels in patients with *P. vivax* malaria. These patients displayed decreased thioredoxin reductase (TrxR) activity, which catalyzes the disulfide reduction of the protein binding site using NADPH, being essential for the maintenance of the redox balance. NO, derived from parasite–host interactions, is involved in the regulation of TrxR expression, which can lead to decreased levels of this enzyme in patients with *P. vivax* malaria [135]. Moreover, it was found that, during *P**. falciparum* infection, the GSH system reduces Trx, which is the substrate for TrxR [130]. On the other hand, GR activity was increased in these patients. GR is involved in maintaining an intracellular reduction environment, as it reduces oxidized glutathione disulfide (GSSG) to the GSH sulfhydryl form, which is crucial for the cell in its defense against oxidative stress. Thus, increased levels of GR may be playing a role in neutralizing the increase in oxidative species and in maintaining homeostasis [136].

The other line of defense consists of antioxidants and cofactors with low molecular weight in the diet, including flavonoids, alpha-tocopherol (vitamin E), ascorbic acid (vitamin C), carotenoids, lipoic acid, selenium, copper, iron, zinc, GSH, and coenzyme Q, among several others [137,138]. These antioxidants work by inactivating ROS/RNS and their derivatives, preventing structural damage, and exhibiting an important protective role during infection. This group of molecules are mainly from exogenous origin and are commonly called small molecules; they can also be classified as consumable antioxidants, since they exist at certain levels within cells, and, because they are not synthetized endogenously, they are consumed when oxidative stress occurs, while enzymatic antioxidants respond with increased activity during oxidative aggression and, therefore, can be classified as mobilizable antioxidants. Along with antioxidant enzymes, other antioxidants can be included in the former group, as they can be synthetized endogenously as a response to oxidative stress. Among these molecules, the most important is GSH. 

The levels of non-enzymatic antioxidants and the total antioxidant status (TAS) in patients with malaria and dengue were evaluated, and an increase in GSH levels was observed in dengue and *P. falciparum* malaria; however, there was a reduction in ascorbate levels in both cases [139]. In another study, the levels of GSH and ascorbic acid were reduced, being associated with increased MDA in patients with uncomplicated *P. falciparum* malaria [34].

Dietary antioxidants were linked to the modulation of host susceptibility or resistance to infectious pathogens. Several pieces of evidence relate micronutrient deficiencies to malaria incidence, especially in endemic areas [140,141]. Malaria itself has been linked to malnutrition, and it is possible that micronutrient deficiencies, including iron (Fe), zinc (Zn), copper (Cu), and vitamin A, can predispose people to malaria [141,142,143].

Micronutrients are antioxidants that play a vital role in fighting anemia and other adverse effects of malaria infection. However, a reduction in total antioxidant capacity is associated with the severity of malaria: the more severe the malaria, the lower the total antioxidant capacity [143].

In assessing the impact of malaria on the total antioxidant status of children, studies have found a strong and negative correlation between parasitemia and total antioxidant levels. In this sense, Cu, Fe, vitamin C, albumin, and β-carotene, which have been reported to have a modulating effect on malaria pathogenesis, were considered deficient in children infected with *Plasmodium* sp. [144,145]. These studies suggest that the reduction of micronutrients may result from reduced intake, as well as from increased endogenous consumption of antioxidants, as a consequence of the oxidative stress associated with malaria infection.

Malaria appears to interfere with children’s nutritional status, with a marked reduction in the speed of linear growth and the impairment of other indices associated with chronic malnutrition. According to Alexandre et al. [146], children who suffer one or more episodes of malaria have a significant negative effect on the speed of linear growth, especially among children between 5 and 10 years of age.

A study by Sakwe et al. [147] also demonstrated an association between malaria and malnutrition. According to the authors, the relationship between malaria and nutritional status was a two-way association. Malnourished children were 2.07 times more likely to be infected by the malaria parasite compared to well-nourished children. On the other hand, malaria-positive children were 1.89 times more likely to be malnourished than uninfected children. This implies that malaria can cause malnutrition, while malnutrition can exacerbate the disease. Other studies suggest that the lack of micronutrients, such as vitamin A, is a mechanism that can explain the effect of malaria on nutritional status [148].

In this sense, vitamin A is essential for normal immune function and can influence the antibody response and cell-mediated immunity against different infectious diseases [20,149]. A study of preschool children showed that the levels of antioxidants A, C, and E in children infected by *Plasmodium* were low, and that the more severe the infection, the lower the levels of these antioxidants [150]. Recent studies show that vitamin deficiency can significantly contribute to the increase in disease morbidity and mortality [151,152].

The low levels of antioxidants observed in these studies may result from the greater use of plasma host antioxidants by *Plasmodium* to neutralize oxidative damage associated with malaria infection. Thus, adequate medical and nutritional management must be ensured to prevent the adverse effects of malaria infection [153].

## 3. Antioxidant Therapy in Malaria

To improve the host’s antioxidant and immunological response against malaria, recent studies have demonstrated the potential benefits of supplementation with antioxidant compounds from plants, mushrooms, vitamins, and drugs, which would act by modulating the host’s response, strengthening the endogenous antioxidant defense against oxidative stress or acting indirectly towards parasite destruction.

Some promising plants tested displayed high antimalarial and/or antioxidant activity, such as the stem bark of *Terminalia albida* [154], which increased the survival rate of mice with cerebral malaria by reducing ROS/RNS; *Terminalia avicennioides*, rich in gallic acid [155]; extracts of leaves, fruits and peels of *Acacia nilotica*, which displayed in vitro schizonticidal activity against *P. falciparum* [156]; flavonoids of *Artemisia annua* L combined with artemisinin [157,158]; flavonoids of *Dacryodes edulis* and *Ficus exasperata* [159]; phenolic and flavonoid compounds present in *Momordica charantia*, *Bixa orellana, Allamanda cathartica, Ficus exasperata; Cymbopogon citratus* [160], *Dissotis rotundifolia* [161], *Trichilia heudelotii, Polyalthia longifolia,* and *Markhamia tomentosa* [162]. It has also been suggested that essential oils from plants present parasiticidal and/or antioxidant activity [163,164].

Brandão et al. [165] identified that the ethanolic extract and alkaloid fraction of *Aspidosperma nitidum*, an Amazonian medicinal plant, reduced parasitemia by 80% on the 5th day of *P. berghei*-infection in mice; however, this reduction was not sustained until the 8th day of infection. In addition, the antiplasmodial activity of the ethanolic extract of *A. nitidum* was identified in vitro against a chloroquine-resistant *P. falciparum* clone. In a similar study conducted with a plant of the Amazon region, Vale et al. [166] identified an in vitro antimalarial action of eleutherin and isoeleutherin naphthoquinones isolated from *Eleutherine plicata*, and this effect was associated with the cytochrome bc 1 complex, according to a molecular modeling study. Such in vitro antimalarial action of quinones isolated from medicinal plants has already been reported as possible candidates for new antimalarials [167,168].

Another therapeutic alternative with possible antimalarial action tested is supplementation with commercial drugs that contain antioxidants in their formulation, such as L-glutathione, NAC, melatonin, fenozyme, deferoxamine, or folate. Among the studied drugs, one that displayed very promising results was NAC, a drug widely used as a mucolytic agent and that has become the target of studies as an antioxidant supplement for adjuvant therapy for malaria, as it is an exogenous precursor of GSH, one of the main components of the antioxidant defense system [27,169,170]. Additionally, Gomes et al. [14], when evaluating the effect of supplementation with NAC or *Agaricus sylvaticus* in *P. berghei*-infected mice, concluded that both supplements reduced parasitemia during infection and increased the total antioxidant capacity of animals, with *A. sylvaticus* being more effective in reducing oxidative damage. Notwithstanding, both represent important adjuvant treatment strategies in malaria. Nevertheless, in another study, NAC was unable to protect pregnant *P. chabaudi*-infected mice [33].

Regarding the action of melatonin, an endogenous hormone synthesized and released by the pineal gland, studies suggest that it can perform important functions in the pathophysiology of malaria [171]. In fact, numerous studies indicate that host melatonin is capable of modulating the intraerythrocytic cycle of the malaria parasite by activating signaling cascades, generating inositol triphosphate (IP3), leading to the release of intracellular calcium, which induces the increase of cyclic adenosine monophosphate (cAMP) and activation of proteins, including the nuclear protein PfMORC, protein kinase eIK1 (PfeIK1), and protein kinase 7 (PfPK7) of *Plasmodium falciparum*, essential for the synchronization of the parasite [172,173,174].

Melatonin has also been described as capable of modulating genes that are coupled to aerobic respiration and ATP production, including S-adenosylmethionine decarboxylase/ornithine decarboxylase, glutamine synthesis, and the synthesis of the alpha subunit of succinyl-CoA and acyl-CoA, favoring the metabolic pathways of the replication of the parasite [175].

Additionally, it has been demonstrated that melatonin is a powerful antioxidant, capable of direct and indirect elimination of ROS/RNS and that it easily crosses the blood–brain barrier, conferring protection to brain cells and providing increased survival to *Plasmodium*-infected animals [176,177,178,179].

In this sense, Ataide et al. [180] showed that melatonin prevented brain damage and neurocognitive changes, thereby improving the survival rate of mice infected with *Plasmodium berghei* ANKA, as well as clinical changes and neurocognitive dysfunction, resulting in protection against changes induced by experimental cerebral malaria. Thus, the neuroprotection of melatonin may be associated with its antioxidant effect that attenuates oxidative stress and the inflammatory process, which are marked features of cerebral malaria [181,182].

Another antioxidant with beneficial effects is fenozyme, a mimetic nanoenzyme of catalase. In the study by Zhao et al. [183], the administration of fenozyme improved the survival rate of animals induced to experimental cerebral malaria, protected the endothelial cells of the blood–brain barrier from ROS/RNS damage, and decreased parasitemia. Additionally, concomitant administration of fenozyme and artemeter improved brain inflammation and memory impairment in mice. These results demonstrated the importance of ROS/RNS in the development of cerebral malaria and indicated that the combination of fenozyme with antimalarial drugs may provide a new treatment strategy.

Another molecule that plays a key role in brain function, energy metabolism, and neurotransmission is iron [184], which is an essential metal for the growth and development of *Plasmodium* [185]. Clinical and epidemiological evidence indicates that iron deficiency in the host protects against malaria and that iron bioavailability can influence the clinical evolution of malaria infection [186,187]. It has been shown that iron chelators, such as deferoxamine and (DFO), can act as effective antimalarial agents due to their ability to interact with iron, leading to iron deprivation by intracellular protozoa and, consequently, inhibiting the growth of the parasite [188]. 

In a recent study, Tiwari et al. [189], reported the potent antimalarial activity of covalent conjugates of natural and artificial iron chelators, such as DFO, ferricrocin, and ICL-670, combined with 1,2,4-trioxolan (ozonide) antimalarials against in vitro drug-resistant and drug-sensitive *Plasmodium falciparum* strains. According to the authors, the combination of iron chelator antioxidants and antimalarial agents is a promising strategy for the prevention of cognitive damage in patients with severe malaria.

According to Dey et al. [190], deferoxamine (300 mg/kg) and NAC (250 mg/kg), can interact with free heme, blocking iron-dependent ROS/RNS reactions and preventing the oxidative stress associated with malaria.

Notwithstanding, it is noteworthy that malaria and malnutrition often coexist, and evidence suggests that malnutrition has a detrimental effect on malaria immunity, especially in pregnant women and children, so that supplementation with nutrients, including iron and folic acid, can help in the development of malaria immunity and reduce malaria susceptibility in high-risk groups [191,192].

Prado et al. [193], in a randomized controlled study of maternal cognition, identified that children of pregnant women who received an iron and folic acid tablet daily had higher visuospatial scores, equivalent to 3 IQ points, compared to those whose mothers received other treatments. 

On the other hand, in a prospective cohort study that included 636 mothers and only children of 828 pregnant women who received daily iron and folic acid supplements throughout pregnancy, Mireku et al. [194] observed that iron deficiency in pregnancy, in the context of iron supplementation, is not associated with serum umbilical ferritin concentration, nor with child cognitive and motor development. 

In fact, iron supplementation does not seem to be completely safe, especially in endemic environments, and may increase susceptibility to malaria [195,196]. In that sense, Brabin et al. [197], in a double-blind study in an endemic malaria area with nulliparaus women who received iron and folic acid or folic acid alone for 18 months, identified that the supplementation of iron to prevent anemia may increase the risk of lower genital tract infections, since the virulence of some pathogens such as *Plasmodium* depends on the availability of iron.

Moreover, a double-blind, randomized, and controlled study of non-inferiority in rural Burkina Faso, Gies et al. [198], in an area of high malaria transmission, found that weekly iron and folic acid supplementation for up to 18 months did not significantly affect the prevalence of *Plasmodium* infection, iron deficiency, or anemia, compared to folic acid supplements isolated in the pregnant or non-pregnant cohort. According to the authors, iron supplementation, as routinely given to populations in a hazardous area, is not useful and is potentially harmful.

Nevertheless, several antioxidants are promising as adjuvant therapy of malaria. In a study by Penna-Coutinho and Aguiar [199], an antimalarial effect was observed both in vivo and in vitro for the drugs Accuvit, Soyfit, and Ginkgo, since they contain antioxidants such as beta-carotene, riboflavin, and flavonoid hesperidin. Another promising flavonoid is quercetin, which presented a protective effect in reducing chloroquine-induced oxidative stress and hepatotoxicity, and the authors proposed combining this antioxidant with standard malaria treatment [200].

Among the studies of antioxidants tested in malaria, the most studied to date are vitamins, which can reduce or prevent oxidative damage generated by antimalarials or triggered by the presence of the parasite in the host.

### 3.1. Antioxidant Vitamin Supplementation in Malaria

Vitamins are responsible for optimizing molecular performance and cellular protection against oxidizing agents, and the intake of these micronutrients is essential for the proper functioning of the immune system. Supplementation with antioxidants for malaria patients represents a complementary alternative to drug treatment, since vitamins promote a better balance between pro-and anti-inflammatory cytokines, which is essential for malaria control [201,202,203].

Vitamin supplementation as an adjuvant to antimalarial therapy is addressed in several studies, both in isolated form and in vitamin complexes, with promising results. Olofin et al. [19] investigated multivitamin supplementation (vitamins B, C, and E) and vitamin A supplementation in the incidence of malaria in women infected with the human immunodeficiency virus (HIV) in reproductive age. The authors noted that multivitamin supplementation significantly reduced the risk of developing clinical malaria, although there was an increase in malaria parasitemia. Vitamin A supplementation did not change the incidence of malaria in this study.

Ojesele et al. [204,205], who tested the co-administration of *Phyllanthus amarus* seed, chloroquine, and artesunate with vitamins A, B, C, and E, pointed to a greater antioxidant capacity and parasiticidal effect of *P. amarus* when combined with vitamins A, B, or E. In addition, there was an increase in the antimalarial activity of drugs combined with vitamins A, B, or E. In this study, supplementation with vitamin C did not show good results.

Few studies have been conducted evaluating the role of vitamin C alone. Despite its known antioxidant capacity, vitamin C supplementation has not displayed relevant results in the treatment of malaria, such as those presented by Ekeh et al. [17], who evaluated the effect of different combinations of vitamin C and Zn on the hematological parameters and mortality of *P. berghei*-infected mice, and found that the supplement improved hematological parameters; however, neither mortality nor parasitemia were significantly reduced.

In contrast, Qin et al. [23], when evaluating the effect of vitamin C supplementation on the immune response to infection by *P. yoelii* in mice, observed that administering vitamin C (25 mg/kg/day, 250 mg/kg/day) increased the immune response during the early stages of malaria infection, regardless of dose. In addition to vitamin C, vitamin A and vitamin D have been studied alone as adjuvants for malaria therapy.

Another important vitamin in antioxidant defense is vitamin A, which acts in modulating innate and adaptive immune responses and protecting cells against the deleterious effects of oxidative stress [20,141]. In this sense, randomized clinical trials have administered vitamin A supplementation to control *Plasmodium* parasitemia. Hamzah et al. [206] evaluated the in vivo efficacy of retinol administration and its effect on lipid peroxidation in a murine model of *P. berghei*. The results showed a prophylactic function of retinol, with the reduction of parasitemia due to the increase of retinol stock within tissues.

Vitamin A supplementation was also evaluated in a murine model of cerebral malaria; however, although no difference was observed in the conditions associated with cerebral malaria, mortality was reduced during adjuvant treatment with this vitamin [207].

Following the principle of the high antioxidant power of vitamin A, several studies associated this vitamin with Zn to enhance cellular protection against oxidative stress generated during malaria infection. Zeba et al. [28], through a randomized, double-blind trial, supplemented children aged 6 years for 72 months with a single dose of 200,000 IU of vitamin A associated with daily Zn supplementation for six months. At the end of the study, there was a significant decrease in the prevalence of malaria and a reduction in the risk of fever and clinical episodes of malaria among supplemented children.

Additionally, Owusu-Agyei et al. [18] evaluated the effect of vitamin A alone and compared it to vitamin A supplementation associated with Zn on the incidence of clinical malaria. They observed a reduction in clinical attacks in children diagnosed with uncomplicated malaria and supplemented with the combination of vitamin A and Zn, when compared to children who received only vitamin A. Similar results were found by Darling et al. [29] who, when supplementing daily doses of vitamin A and Zn orally with one or both nutrients in the first trimester of pregnancy to assess possible reductions in the risk of placental malaria and adverse pregnancy outcomes, observed that supplementation with 25 mg/day was associated with a 36% reduction in the risk of positive histopathological placental infection. However, vitamin A supplementation had no impact on placental malaria.

In a more recent study, Chen et al. [21] investigated the immunoregulatory effects of vitamin A supplementation in mice infected with *P. yoelii* or *P. berghei*, observing a reduction in parasitemia. Nevertheless, the mortality rate was not significantly altered in animals infected with *P. yoelii*, while, in those infected with *P. berghei,* the treatment reduced parasitemia and delayed the outcome of cerebral malaria. These data corroborate the effectiveness of vitamin A as an adjuvant supplement in the treatment of malaria, in addition to positively influencing the reduction of mortality.

Another vitamin studied for its antioxidant properties is vitamin D, a fat-soluble vitamin synthesized in the skin after exposure to solar ultraviolet radiation or provided by the diet. In addition to its traditionally known roles in the regulation of bone metabolism and calcium–phosphorus homeostasis, vitamin D is able to act through several mechanisms, including protein expression or the modulation of oxidative stress, inflammation, and cellular metabolism [208]. Nevertheless, only a few vitamin D supplementation studies have been developed so far, although it is believed to be very beneficial as an adjuvant therapy for malaria. Nevertheless, more clinical trials need to be carried out to confirm this hypothesis [152].

He et al. [24], showed that oral vitamin D supplementation protected the occurrence of cerebral malaria during infection by *P. berghei* in susceptible mice. The authors suggested the protective effect of vitamin D occurred through the inhibition of a strong proinflammatory response from the host (IFN-γ and TNF-α), mediated directly by its action on Th1 cells, as well as indirectly through the inhibition of innate immunity. In addition, treatment with vitamin D expanded the regulatory T cells (Tregs) and inhibited the differentiation, maturation, and functioning of DC, which altogether resulted in the increased expression of IL10.

The antimalarial activity in knockout mice for the vitamin D3 receptor (VDR) and its 22-oxacalcitriol analog (22-OCT), which causes less hypercalcemia than vitamin D3, is related to the direct and indirect action of VDR, resulting in a reduction of IFN-γ with a consequent increase in the survival of infected animals [209].

In the study by Dwivedi et al. [15], the combined administration of artemether/lumefantrine and vitamin D significantly improved the survival of C57BL mice infected with *P. berghei* and conferred protection against cerebral malaria. In addition, the integrity of the blood–brain barrier was restored. Recently, Wu et al. [25] also verified the preventive role of vitamin D in an experimental model of cerebral malaria in mice. The oral administration of vitamin D improved the inducible inflammatory responses, reducing IFN-γ and TNF-α, and decreasing the expression of these cytokines in spleen cells, as well as decreasing the expression of the mRNA of the chemokines CXCL-9 and CXCL-10 in brain cells, suggesting that the prophylactic oral administration of vitamin D, through a multifactorial process, results in the maintenance of the blood–brain barrier and improves animal survival.

Another fat-soluble vitamin tested is vitamin E. However, few studies have shown beneficial effects from supplementing this vitamin in the treatment of malaria. In a study developed by Ibrahim et al. [210], the intraperitoneal administration of vitamin E prevented anemia, changes in the weight of the animals’ organs, and reversed the oxidative changes caused by *P. berghei*-infection. In the same year, Ibrahim et al. [211] found the same results, but using the combination of vitamins E and C. However, in a previous study, the oral administration of vitamin E reduced the antimalarial action of artesunate and delayed the improvement in the hematocrit of animals infected with *P. berghei*, showing an antagonistic effect of vitamin E when combined with the medication [212]. It has been proven that *P. falciparum* synthesizes vitamin E for its own antioxidant defense as a protection against ROS generated by antimalarial treatment, but when vitamin E biosynthesis is inhibited in the parasite by succinic acid, there is an increase in ROS levels, causing the death of the parasite [213].

Studies showed that a reduction of vitamin E has a beneficial and protective effect on malaria and that this effect is related to the inhibition of the transfer protein α-tocopherol (α-TTP), which regulates the levels of this vitamin. However, vitamin E deficiency in the diet impairs the integrity of the erythrocyte membrane that is susceptible to ROS/RNS aggression [26,214,215].

### 3.2. Antimalarial Therapy Combined with Antioxidant Supplementation

The parasite is sensitive to in vitro and in vivo oxidative stress induced by antimalarials, such as artemisinin [77], chloroquine [216], and primaquine [72] which are sources that generate free radicals, causing the death of the parasite by redox imbalance [213]. 

According to WHO recommendations, artemisinin is the main drug that can be combined with other antimalarials, such as artemether-lumefantrine, artesunate-amodiaquine, dihydroartemisinin-piperaquine, artesunate-sulfadoxine-pyrimethamine, artesunate-pyronaridine, and artesunate-mefloquine. 

Artemisinin is identified by the presence of the endoperoxide group in its structure, which is responsible for its antimalarial activity [217], for it alkyls the heme group and prevents its polymerization in hemozoin, causing an increase in ROS/RNS production [218]. However, long-term resistance to the combination of artemisinin with these drugs has been reported [219,220].

To reduce the damage caused by ROS/RNS as a consequence of infection and antimalarial treatment, supplementation with antioxidants has been suggested. However, according to Isah and Ibrahim [27], the role of antioxidant therapy in malaria is controversial, since antimalarials act by inducing oxidative stress to destroy parasites and since antioxidants could interfere in the pharmacological action of these drugs. Notwithstanding, among the antioxidants, the most studied are NAC, vitamins C and E, and folic acid.

Regarding NAC, the study by Fitri et al. [221] sought to evaluate whether combined artemisinin and NAC therapy was more effective compared to artemisinin monotherapy, and the results showed that this combination reduces MDA levels in lung and brain tissue in experimental malaria caused by *P. berghei* compared to artemisinin monotherapy. In a similar study, it was identified that the combined therapy between chloroquine and NAC results in a synergistic effect, generating a reduction of parasitemia and MDA levels in *P. berghei*-infected mice [222].

Mckoy et al. [223], when evaluating the in vitro effects of artemeter/lumefantrine co-incubation together with vitamin C on blood viscosity and elasticity, identified the potentiation of the hemolytic effects of antimalarial drugs, reducing blood viscosity and elasticity. These results suggest that hemolysis is generated by vitamin C supplementation associated with artemeter/lumefantrine in patients during antimalarial therapy. Based on this, to avoid a possible loss of action of antimalarials, the administration of vitamin C is not recommended in patients with malaria, since this combination impairs the rate of elimination of the parasite. This effect is particularly related to high doses of vitamin C, which may inhibit the growth of *Plasmodium* only to some extent [22]. Thus, vitamin C can exert a pro-oxidant effect depending on its dose, generating an increase in intracellular ROS in red blood cells during the intraerythrocytic phase of the parasite [224]. 

However, in a study conducted by Iyawe and Onigbinde [225], who sought to identify the effect of treatment with chloroquine, folic acid, and ascorbic acid on malaria, it was observed that the combined treatment between chloroquine and folic acid was more effective than ascorbic acid combined with chloroquine in the experimental malaria model in mice. In a recent study, Ebohon et al. [226] identified that the co-administration of vitamin C with oral artesunate-amodiaquine in experimental malaria may be beneficial because it reduced oxidative stress and increased the gene expression of antioxidant enzymes in mice infected by *Plasmodium berghei*. 

## 4. Final Remarks

The involvement of oxidative stress in malaria is complex and progresses with changes in the host’s pro-oxidant and antioxidant balance. The main sources of antioxidant defenses involved in malaria are: (1) host enzyme antioxidants (SOD, CAT, GSH-Px, GST, GR, Prx, and Trx); (2) host non-enzymatic antioxidants (vitamins A, D, E, and C; carotenoids; uric acid; and GSH); (3) non-enzymatic antioxidants from the host’s metal chelation system (iron, copper, and metallothionein-chelating proteins); (4) exogenous antioxidants of dietary origin or drugs (flavonoids, phenolic compounds, minerals, NAC, and the mushroom *A. sylvaticus*).

The studies presented in this revision showed that supplementation, mainly with vitamins A and D, is advantageous. This provides a potential and beneficial strategy for the prophylactic application of vitamins and the prevention of malaria.

Notwithstanding, supplementation with antioxidants from different sources would act in the prevention and control of oxidative changes generated by the parasite during infection, with a consequent increase in the total antioxidant capacity and a reduction of cellular damage to the host, constituting a complementary therapy to antimalarial treatment. However, further studies are needed to corroborate this hypothesis.

## Figures and Tables

**Figure 1 ijms-23-05949-f001:**
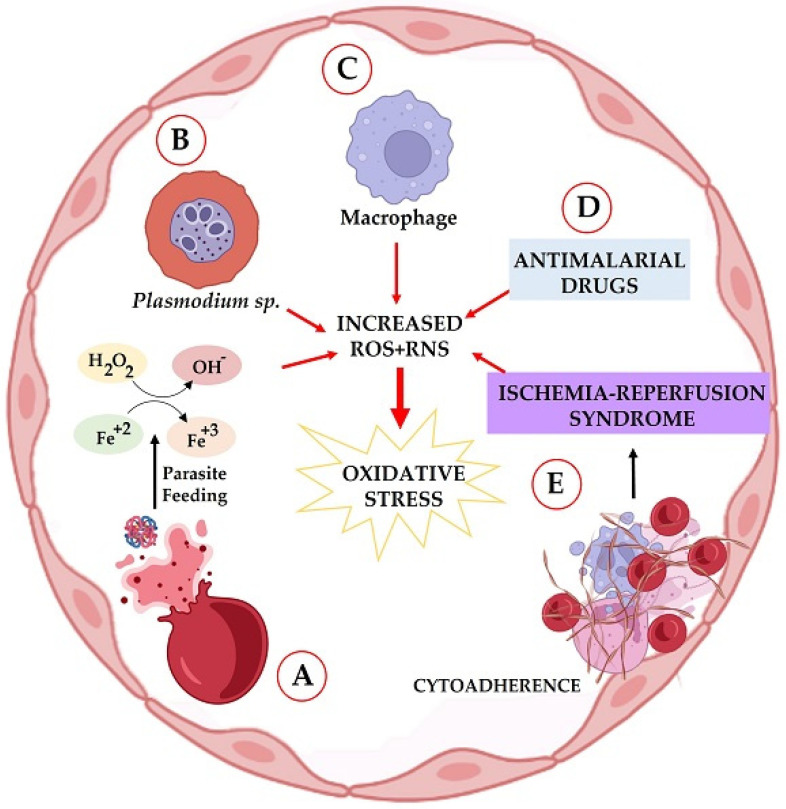
Sources of oxidative stress in malaria. A = Free-iron-induced oxidative stress (OS), as a consequence of hemoglobin digestion by the parasite and through the Fenton or Haber–Weiss reactions; C = direct ROS/RNS production by the parasite; B = host’s inflammatory response-derived oxidative stress; D = production of ROS/RNS by antimalarial drug metabolism; E = cytoadherence- and/or anemia-derived oxidative stress, through ischemia-reperfusion syndrome. ROS = Reactive oxygen species; RNS = reactive nitrogen species.

**Figure 2 ijms-23-05949-f002:**
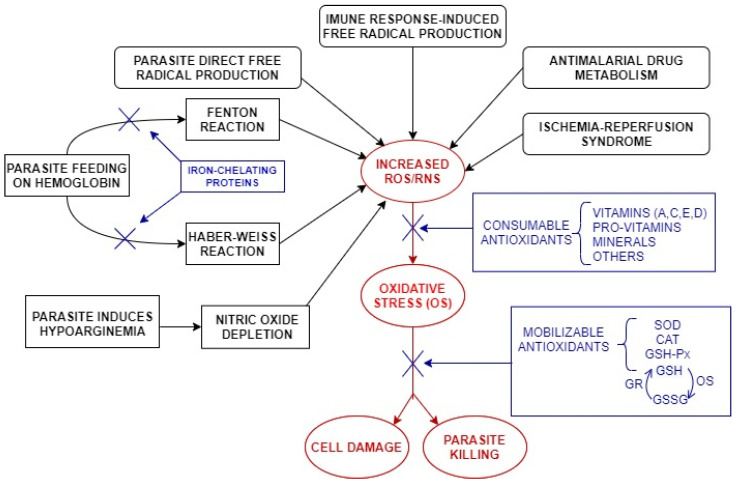
Antioxidant defense (blue) and mechanisms of free radical involvement (black) in malaria. ROS = reactive oxygen species; RNS = reactive nitrogen species; SOD = superoxide dismutase; CAT = catalase; GSH-Px = glutathione peroxidase; GSH = reduced form of glutathione; GR = glutathione reductase; GSSG = oxidized form of glutathione.

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
