# Peer review of "Oxidative Stress in Malaria: Potential Benefits of Antioxidant Therapy"

_ijms, 2022, doi:10.3390/ijms23115949_

Round 1
Reviewer 1 Report
Authors need to update global data from the latest reports i.e.
https://www.mmv.org/newsroom/publications/world-malaria-reports
with estimated projections in coming years from
https://www.who.int/data/gho/data/themes/malaria
Authors need to hi critical reference of treatment of placental malaria by antioxidant therapy and saving the pregnancy in the introduction section which is a clinically proven mechanism.
The arginine therapy is controversial as the parasite has highly active arginase which causes hypoarginemia in the host. The parasite cannot consume citrulline which has a better treatment potential as a NO donor.
Hypoarginemia caused by malaria and resulting NO depletion needs to be added in Figure2.
Authors need to add a section on the contribution of approved and widely used antimalarials and antipyretics on ROS.. e.g. artemisinin. Also, possible effect of antioxidant therapy on the antimalarial efficacy of these drugs needs to be discussed as many generate ROS in parasites which is a crucial part of their antiparasitic activity.
Enzymatic ROS scavenger mediated protection from cerebral malaria e.g Fenozyme
Author Response
Comments and Suggestions – Reviewer 1
Dear reviewer,
On behalf of all authors, I thank you for your consideration of our manuscript, and for the comprehensive review, which shall result in a considerable improvement of it. We believe to have addressed all points raised in your review, to which we have responded in a point-by-point fashion, with the best of our capacity. All amendments in the text were highlighted in blue.
Best regards,
Sandro Percario, DSc, PhD
- Authors need to update global data from the latest reports i.e.
https://www.mmv.org/newsroom/publications/world-malaria-reports
with estimated projections in coming years from
https://www.who.int/data/gho/data/themes/malaria.
R- Data was update to the last version of the Malaria Report (2021), and reference was corrected.
- Authors need to hi critical reference of treatment of placental malaria by antioxidant therapy and saving the pregnancy in the introduction section which is a clinically proven mechanism.
R- Information was added at the end of the introduction.
- The arginine therapy is controversial as the parasite has highly active arginase which causes hypoarginemia in the host. The parasite cannot consume citrulline which has a better treatment potential as a NO donor.
Hypoarginemia caused by malaria and resulting NO depletion needs to be added in Figure2.
R- Information was added at the end of section 2.1, with 7 new references cited. The amendment in figure 2 was performed.
- Authors need to add a section on the contribution of approved and widely used antimalarials and antipyretics on ROS.. e.g. artemisinin. Also, possible effect of antioxidant therapy on the antimalarial efficacy of these drugs needs to be discussed as many generate ROS in parasites which is a crucial part of their antiparasitic activity.
R- The purpose of this revision is to update the discussion on the potential beneficial effects of antioxidant-alone or -combined with antimalarials in the treatment of malaria, that was raised in our previous publication of ten year ago (doi:10.3390/ijms131216346). The inclusion of specific effects of each antioxidant on the efficacy of antimalarials or highlight underlying mechanisms of action of antimalarials may cause a significant increase in manuscripts length, considering that this update has already consumed more than 30 pages and over 200 references. Therefore, we decided to add only a generic statement is this regard.
- Enzymatic ROS scavenger mediated protection from cerebral malaria e.g Fenozyme.
R- Information was added within section 3.
Reviewer 2 Report
The manuscript is a descriptive review of the literature focusing on the role of oxidative stress in malaria and the potential use of antioxidants supplementation as an adjuvant antimalarial treatment. Subject matter is of interest; however, several general issues have been already reported in a similar article published by the authors in 2012 (doi:10.3390/ijms131216346). In my opinion the manuscript should be reorganized and revised by giving greater prominence to pharmacology both in terms of classical drugs used in malaria therapy and antioxidants supplementation.
Specific comments:
Antioxidants that act strictly as free radical scavengers may ameliorate plasmodium-induced organ pathology and cellular damage but unfortunately tend to improve the survival of the mature parasite in the host. Hence, the choice of which antioxidant to use in malaria therapy
depends on which aspect of malarial pathology one wishes to target. Furthermore, some antioxidants interfere with the action of antimalarial drugs. These issues should be reported and discussed in detail.
Beside Vitamins and natural- derived compounds, several antioxidant molecules have been found to be beneficial in malaria disease, among these NAC, folate, melatonin and Deferoxamine. These agents should be added as well as the related experimental and clinical findings.
English language should be revised
Author Response
Comments and Suggestions - Reviewer 2
Dear reviewer,
On behalf of all authors, I thank you for your consideration of our manuscript, and for the comprehensive review, which shall result in a considerable improvement of it. We believe to have addressed all points raised in your review, to which we have responded in a point-by-point fashion, with the best of our capacity. All amendments in the text were highlighted in blue.
Best regards,
Sandro Percario, DSc, PhD
The manuscript is a descriptive review of the literature focusing on the role of oxidative stress in malaria and the potential use of antioxidants supplementation as an adjuvant antimalarial treatment. Subject matter is of interest; however, several general issues have been already reported in a similar article published by the authors in 2012 (doi:10.3390/ijms131216346).
- In my opinion the manuscript should be reorganized and revised by giving greater prominence to pharmacology both in terms of classical drugs used in malaria therapy and antioxidants supplementation.
R- The purpose of this revision is to update the discussion on the potential beneficial effects of antioxidant-alone or -combined with antimalarials in the treatment of malaria, that was raised in our previous publication of ten year ago. To include specific effects or highlight underlying mechanisms of action of antimalarials may cause a significant increase in manuscripts length, considering that this update has already consumed more than 30 pages and over 200 references. Therefore, we decided not to comply with this suggestion. Nevertheless, we included section 3.2 - Antimalarial therapy combined with antioxidant supplementation.
Specific comments:
- Antioxidants that act strictly as free radical scavengers may ameliorate plasmodium-induced organ pathology and cellular damage but unfortunately tend to improve the survival of the mature parasite in the host. Hence, the choice of which antioxidant to use in malaria therapy depends on which aspect of malarial pathology one wishes to target. Furthermore, some antioxidants interfere with the action of antimalarial drugs. These issues should be reported and discussed in detail.
R- Information in this regard was inserted in multiple parts of the manuscript, such as “…melatonin has also been described as capable of modulating genes that are coupled to aerobic respiration and ATP production, including S-adenosylmethionine decarboxylase/ornithine decarboxylase, glutamine synthesis, synthesis of alpha subunit of succinyl-CoA and acyl-CoA, favoring the metabolic pathways of replication of the parasite…”, “…, iron supplementation does not seem to be completely safe, especially in endemic environments, and may increase susceptibility to malaria…, in a double-blind study in an endemic malaria area with nulliparaus women who received iron and folic acid or folic acid alone for 18 months, identified that the supplementation of iron to prevent anemia may increase the risk of lower genital tract infections, since the virulence of some pathogens such as Plasmodium depends on the availability of iron…”, “It has been proven that P. falciparum synthesizes vitamin E for its own antioxidant defense as a protection against ROS generated by antimalarial treatment, but when vitamin E biosynthesis is inhibited in the parasite by the succinic acid, there is an increase in ROS levels causing the death of the parasite…”, “…the role of antioxidant therapy in malaria is controversial, since antimalarials act by inducing oxidative stress to destroy parasites and that antioxidants could interfere in the pharmacological action of these drugs.”.
- Beside Vitamins and natural- derived compounds, several antioxidant molecules have been found to be beneficial in malaria disease, among these NAC, folate, melatonin and Deferoxamine. These agents should be added as well as the related experimental and clinical findings.
R- Information regarding mentioned antioxidants were added within section 3, with 27 new references cited.
- English language should be revised.
R- The manuscript was re-analyzed by a native English speaker.

Round 2
Reviewer 2 Report
The manuscript had been substantially improved therefore it deserves to be published in IJMS